# MGP-AttTCN: An Interpretable Machine Learning Model for the Prediction of Sepsis

### Abstract

With a mortality rate of 5.4 million lives worldwide every year and a healthcare cost of more than 16 billion dollars in the USA alone, sepsis is one of the leading causes of hospital mortality and an increasing concern in the ageing western world. Recently, medical and technological advances have helped re-define the illness criteria of this disease, which is otherwise poorly understood by the medical society. Together with the rise of widely accessible Electronic Health Records, the advances in data mining and complex nonlinear algorithms are a promising avenue for the early detection of sepsis. This work contributes to the research effort in the field of automated sepsis detection with an open-access labelling of the medical MIMIC-III data set. Moreover, we propose MGP-AttTCN: a joint multitask Gaussian Process and attention-based deep learning model to early predict the occurrence of sepsis in an interpretable manner. We show that our model outperforms the current state-of-the-art and present evidence that different labelling heuristics lead to discrepancies in task difficulty.

## 1 Introduction

Every year, it is estimated that 31.5 million people worldwide contract sepsis. With a mortality rate of 17% in its benign state and 26% for its severe state (Fleischmann et al., 2016), sepsis is one of the leading causes of hospital mortality (Vincent et al., 2014), costing the healthcare system more than 16 billion dollars in the USA alone (Angus et al., 2001). Studies demonstrated that early treatment has a significant positive effect on the survival rate (Kumar et al., 2006; Nguyen et al., 2007). In particular, Castellanos-Ortega et al. (2010) demonstrated that each hour delay in treating a patient results in a 7.6% increase in mortality.

Current methods of screening, such as Modified Early Warning System (MEWS) and Systemic Inflammatory Response Syndrome (SIRS) have been criticised for their lack of specificity, leading to low accuracies and high false alarm rates. In 2015, the Third International Consensus Definitions for Sepsis (Singer et al., 2016; Seymour et al., 2016; Shankar-Hari et al., 2016) committee worked towards incorporating medical and technological advances into an up-to-date definition of sepsis, providing scientists with widely acknowledged illness criteria. Together with the rise of Electronic Health Records (EHR), the scientific community is now armed with both the data and labelling techniques to experiment with novel prediction methods (Islam et al., 2019; Henry et al., 2015; Ghosh et al., 2017; Calvert et al., 2016; Desautels et al., 2016), which are already proving effective in increasing survival rate (Shimabukuro et al., 2017) and promising in decreasing costs.

New models developed so far either relied on some interpretable yet simple prediction methods, such as logistic regression (Calvert et al., 2016) and decision tree based classifiers (Mao et al., 2018; Delahanty et al., 2019), or on effective yet black-box methods such as Recurrent Neural Networks (Futoma et al., 2017b). Moreover, the results achieved by different authors are rarely comparable: although most use the MIMIC-III data set, the disparities in labelling rules result in highly variable data sets (eg. Raghu et al. (2018) have 17,898 septic patients vs. 2,577 for Desautels et al. (2016)).

This work presents an attempt at reconciling interpretability and predictive performance on the sepsis prediction task and makes the following contributions:

- Gold standard for labelling. We provide a gold standard for Sepsis-3 labelling implemented on the MIMIC-III data set.

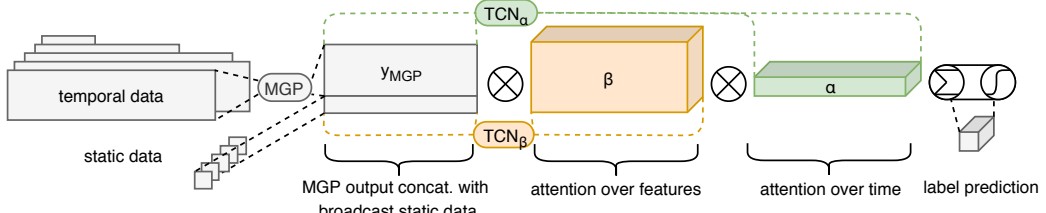

Figure 1: Proposed model architecture.

- Novel interpretable model. We present an explainable and end-to-end trainable model based on Multitask Gaussian Processes and Attentive Neural Networks for the early prediction of sepsis.
- Empirical evaluation. We assess our model on real-world medical data and report superior predictive performance and interpretability compared to previous methods.

An overview of our proposed method is shown in Figure 1.

## 2 RELATED WORK

**Medical time series diagnosis**    Multiple researchers have tackled the task of predicting sepsis and septic shock. Works on septic shock include exploration of survival models (Henry et al., 2015) and Hidden Markov Models (Ghosh et al., 2017). However, these models rely on the assumption that a patient has already developed sepsis and focus on patterns of patients' further deterioration. Other authors (Calvert et al., 2016; Desautels et al., 2016; Mao et al., 2018; Delahanty et al., 2019) use linear models and decision trees on engineered features to predict sepsis onset, thus failing to capture temporal patterns. More recently, Kam & Kim (2017) and Raghu et al. (2018) employed recurrent neural networks to better capture time dependencies. Crucially, all these models rely on either averaging or forward imputation of data points to create equidistant inputs. This transformation creates data artefacts and discards relevant uncertainty: in the medical field, the absence of data is a conscious decision made by professionals implying an underlying belief of the patient state. Futoma et al. (2017b) and Moor et al. (2019) tackled this issue with Multitask Gaussian Processes (MGPs), however their models lack the interpretability necessary in the medical field.

**Irregularly sampled time series**    The most common solution to missing values is forward imputation (Calvert et al., 2016). Lipton et al. (2016) utilise forward imputation coupled with a missingness indicator fed into a black-box model. Although this method retains information about data presence, it is not clear how the information is later interpreted by the model and hence does not meet our transparency criteria. Ghassemi et al. (2015) use MGPs to fit sparse medical data, however they optimise their model for the data fit and use the parametrisation as input for a classifier rather than optimising the model for a classification task. Both Futoma et al. (2017a) and Moor et al. (2019) use MGPs with end-to-end training, although their temporal covariance function is shared across all variables. Finally, Futoma et al. (2017b) uses MGPs with multiple time kernels in a similar fashion to our model, although they infer the number of kernels from hyperparameter tuning rather than the data itself.

**Attention based neural networks**    Attention was first introduced on the topic of machine translation (Bahdanau et al., 2014). Since then, the concept has been used in natural language processing (Yang et al., 2016; Yu et al., 2018) and image analysis (Mnih et al., 2014; Schlemper et al., 2019). In the same spirit, Qin et al. (2017) used attention mechanisms to improve the performance of a time series prediction model. Although their model easily explains the variable importance, its attention mechanism is based on Long Short Term Memory encodings of the time series. At any given time, such an encoding contains both the information of the current time point and all previous time points seen by the recurrent model. As such, the time domain attention does not allow for easy interpretation. More similar to our implementation is the RETAIN model (Choi et al., 2016), which generates its attention weights through reversed recurrent networks and applies them to a simple embedding of the

time series. The model employs recurrent neural networks which are slower to train and suffer from the vanishing gradient problem. Furthermore, the initial and final embeddings decrease the model's interpretablity. Other authors using Temporal Convolutional Network (TCN) based attention include Lin et al. (2019), who only attend to time.

## 3 METHOD

### 3.1 NOTATION

Let us first define some notation for the problem at hand. For each patient encounter $p$, several features $y_{p,t_i,k}$ are recorded at times $t_{p,k,i}$ from admission, where $k \in \{1, \ldots, M\}$ is the feature identifier. These features are often vital signs and laboratory results. As such, they are rarely observed at the same times. Hence, we have a sparse matrix representation of observations

$$\begin{pmatrix} y_{p,1,t_1} & \cdots & y_{p,1,t_{N_p}} \\ \vdots & \ddots & \\ y_{p,M,t_1} & \cdots & y_{p,M,t_{N_p}} \end{pmatrix} \tag{1}$$

where $N_p$ is the patient's observation period length. We also define static features $\mathbf{s}_p = \{s_{p,M+1}, ..s_{p,M+Q}\}$ with features identifiers $k \in \{M+1, \ldots, M+Q\}$, corresponding to time-independent quantities, such as age, gender and first admission unit. Finally, we define sepsis labels $l_p \in \{0,1\}$. Given the sparsity of the data, we can define the compact representation of all observed values:

$$\{\mathbf{t}_p, \mathbf{y}_p, \mathbf{s}_p, l_p\} = \left\{ \{t_{p,i,k}, y_{p,i,k}\}_{i \in \{0,\ldots,N_p\}, k \in \{1,\ldots,M\}}, \quad \{s_{p,M+1}, ..s_{p,M+Q}\}, \quad l_p \right\} \tag{2}$$

The goal of the model is, for a given set $\{\mathbf{t}_p, \mathbf{y}_p, \mathbf{s}_p\}$ to predict the label $l_p$. In order to remove clutter, we will from now on drop the patient-specific subscript $p$ from all notation, and the feature subscript $k$ from time notation, simplifying $t_{p,k,i}$ to $t_i$.

### 3.2 MULTITASK GAUSSIAN PROCESS (MGP)

Gaussian processes are commonly known for their ability to generate coherent function fits to a set of irregular samples, by modelling the data covariance. As they easily account for uncertainty and do not require homogeneously sampled data, Gaussian processes are the perfect candidate model to deal with relatively small amounts of medical data.

Following Bonilla et al. (2008), we use a Multitask Gaussian Process (MGP) to capture feature correlation and Li & Marlin (2016)'s end-to-end training framework, in a similar manner to Futoma et al. (2017a). Given an hourly spaced time series $\{t'_i\}_{i=-N_p}^0$ (where 0 is the time of prediction), the MGP layer produces a set of posterior predictions for each feature, which will then be fed into a classification model.

We define a patient-independent prior over the true values of $\{y_{i,k}\}$ by $\{f_k(t_i)\}$ such that

$$\{y_{i,k}\} \sim \mathcal{N}(f_k(t_i), \sigma_k^2) \tag{3}$$

$$\langle f_k(t_i), f_{k'}(t_j) \rangle = \sum_{l \in L} K_l^{\mathbf{k}}(k, k') \, K_l^{\mathbf{tt}}(t_i, t_j) \tag{4}$$

where $\{K_l^{\mathbf{tt}}(t_i, t_j)\}_{l \in L}$ are time point covariances varying in smoothness, $\{K_l^{\mathbf{k}}(k, k')\}_{l \in L}$ are feature covariances at a given smoothness level, independent of time, and $L$ are smoothness clusters. Over all variables and time points, the multivariate model has covariance

$$\sum_{l \in L} K_l^{\mathbf{k}} \otimes K_l^{\mathbf{tt}} + D \otimes I \tag{5}$$

where $D = diag(\sigma_k)$ are the noise terms associated to each feature and $\otimes$ is the Kronecker product. The posterior over $\mathbf{t}' = \{t'_i\}_{i=-N_p}^0$ is a multivariate Gaussian with mean and covariance:

$$\begin{aligned} \boldsymbol{\mu} &= \big( \sum_{l \in L} K_l^{\mathbf{k}} \otimes K_l^{\mathbf{tt'}} \big) \big( \sum_{l \in L} K_l^{\mathbf{k}} \otimes K_l^{\mathbf{tt}} + D \otimes I \big)^{-1} \mathbf{y} \\ \Sigma &= \sum_{l \in L} K_l^{\mathbf{k}} \otimes K_l^{\mathbf{t't'}} - \big( \sum_{l \in L} K_l^{\mathbf{k}} \otimes K_l^{\mathbf{tt'}} \big) \big( \sum_{l \in L} K_l^{\mathbf{k}} \otimes K_l^{\mathbf{tt}} + D \otimes I \big)^{-1} \big( \sum_{l \in L} K_l^{\mathbf{k}} \otimes K_l^{\mathbf{t't}} \big) \end{aligned} \tag{6}$$

In order to approximate the posterior distribution, we then take Monte Carlo samples $\mathbf{y}_{\text{MC}}$ from $\mathbf{Y}_{\text{MGP}} \sim \mathcal{N}(\boldsymbol{\mu}, \Sigma)$.

Note that there are two main feature clusters: vital signs (vitals) and laboratory results (labs). Vitals are noisier and sampled more often, whereas labs are more monotone and rarely sampled. As opposed to Futoma et al. (2017b), we do not treat the number of clusters $L$ as hyperparameters but set $L = 2$ and define

$$K_l^t(t_i, t_j) = \exp\left(\frac{-|t_i - t_j|}{\lambda_l}\right) \tag{7}$$

as Ornstein-Uhlenbeck (OU) kernels with lengths $\lambda_1$ and $\lambda_2$, each representing a cluster smoothness. OU kernels are well suited to capture local variations and do not assume infinite differentiability as Squared Exponential kernels do. In our case, differentiablity implies a level of smoothness which does not apply to medical records and only introduces unnecessary bias. In addition, given the scarce availability of labs, all short lengthscales would be an ill fit to the data. We hence discarded kernels varying over lengthscales such as the Cauchy and the Rational Quadratic kernels. $K_l^k(k, k')$ are free-form covariance matrice that are learned by gradient descent.

To feed the MGP samples into the classifier, we fix the model time window to $N = 25$ by either zero padding or truncating the beginning of the time series. We choose to do so at the beginning of the time series in order to align prediction times as the last step of the temporal classification model. Here, we also integrate the static variables by broadcasting them over each time point[1].

## 3.3 ATTENTION TIME CONVOLUTIONAL NETWORK (ATTTCN)

The concept of attention was born in machine translation (Bahdanau et al., 2014): given an input sentence embedding $S = \{\mathbf{h}_1, \dots \mathbf{h}_{|S|}\}$, the attention mechanism produces weights $\{\alpha_1^i, \dots \alpha_{|S|}^i\}$ such that $\alpha_j^i \in [0, 1]$, $\sum_j \alpha_j^i = 1$, and a context vector $\mathbf{c}_i = \sum_j \alpha_j^i \mathbf{h}_j$ used to predict target word $i$. The weights $\alpha_j^i$ can therefore be interpreted as the importance of the input sentence's $j^{\text{th}}$ word to produce the $i^{\text{th}}$ word of the translation.

More recently, Choi et al. (2016) have applied attention to clinical time series. Given a time series $\{\mathbf{x}_1, \dots \mathbf{x}_T\} \subset \mathbb{R}^r$, the authors first create a time-independent embedding of the data $\{\mathbf{v}_1, \dots \mathbf{v}_T\} \subset \mathbb{R}^m$. They then use inversed recurrent neural networks (RNN) to create weights $\boldsymbol{\alpha} \in \mathbb{R}^T$ and $\boldsymbol{\beta} \in \mathbb{R}^{T \times m}$, where $\alpha_j \in [0, 1]$ and $\beta_{ij} \in [-1, 1]$, with softmax and tanh activations respectively. The context vectors then take the form $c_i = \sum_{j \leq i} \alpha_j \boldsymbol{\beta}_j \odot v_j$ and are fed into a multilayer perceptron with softmax activation to yield a prediction.

The attention model we devised borrows some ideas from Choi et al. (2016). The interpolated data $\mathbf{y}_{\text{MC}} \in \mathbb{R}^{N \times (M+Q)}$ is directly fed into two temporal convolutional networks (TCNs) (Lea et al. (2017)) and generates embeddings $\mathbf{z} = [z_1, \dots, z_N] \in \mathbb{R}^{N \times (M+Q)}$ and $\mathbf{z}' = [z_1', \dots, z_N'] \in \mathbb{R}^{N \times (M+Q)}$.

TCNs are a class of neural networks composed of *causal* convolutions stacked into Residual Blocks. A causal convolution is a 1D convolutional layer which only takes inputs from the past to generate its output, avoiding any information leakage from the future. Residual Blocks are made of two causal convolutional layers together with ReLU activation functions, dropout and L2 regularisations. The Residual Blocks also include an identity map from the input of the block added to the output. As we only use up to 12 layers, this last step is omitted in our architecture. TCNs have shown to outperform RNNs (Bai et al. (2018)), are faster at training and do not suffer from vanishing gradients. Given the latter, inverting the time series similarly to Lea et al. (2017) also becomes an unnecessary step which we omit.

We generate the attention weights $\alpha$ and $\beta$ as

$$\alpha_{j,0} = \text{softmax}(\mathbf{z}_j \times \mathbf{W}_{\alpha,0} + b_{\alpha,0}) \qquad \alpha_{j,1} = \text{softmax}(\mathbf{z}_j \times \mathbf{W}_{\alpha,1} + b_{\alpha,1}) \tag{8}$$

$$\boldsymbol{\beta}_{j,0} = \text{sigmoid}(\mathbf{z}_j' \times \mathbf{W}_{\beta,0} + \mathbf{b}_{\beta,0}) \qquad \boldsymbol{\beta}_{j,1} = \text{sigmoid}(\mathbf{z}_j' \times \mathbf{W}_{\beta,1} + \mathbf{b}_{\beta,1}) \tag{9}$$

$$\mathbf{W}_{\alpha,0}, \mathbf{W}_{\alpha,1} \in \mathbb{R}^{M+Q} \qquad b_{\alpha,0}, b_{\alpha,1} \in \mathbb{R} \tag{10}$$

$$\mathbf{W}_{\beta,0}, \mathbf{W}_{\beta,1} \in \mathbb{R}^{(M+Q) \times (M+Q)} \qquad \mathbf{b}_{\beta,0}, \mathbf{b}_{\beta,0} \in \mathbb{R}^{M+Q} \tag{11}$$

---

[1]see Appendix C.2 for more information on this design choice

such that $\boldsymbol{\alpha} = [\boldsymbol{\alpha}_0, \boldsymbol{\alpha}_1] \in \mathbb{R}^{N \times 2}$ and $\boldsymbol{\beta} = [\boldsymbol{\beta}_0, \boldsymbol{\beta}_1] \in \mathbb{R}^{N \times (M+Q) \times 2}$.

We then create two context vectors, one for each of negative and positive label predictions

$$\mathbf{c}_i = \sum_{j \leq i} \alpha_{j,\delta} \boldsymbol{\beta}_{j,\delta} \odot \mathbf{y}_{\mathrm{MC},j} \in \mathbb{R}^{N \times (M+Q) \times 2}, \quad \delta \in \{0, 1\} \tag{12}$$

where $\mathbf{y}_{\mathrm{MC},j}$ is broadcast to meet the dimensionality of $\boldsymbol{\beta}_{j,\delta}$. We then predict the labels as

$$\hat{\mathbf{l}}_i = \mathrm{softmax}\Big( \sum_n^N \sum_m^{M+Q} \mathbf{c}_{i,nm} \Big) \in [0, 1]^2 \tag{13}$$

In our case, we are only interested in making predictions with the latest available data. We therefore only use $\hat{\mathbf{l}}_{\mathrm{last}}$ to train the model. This of course can be easily modified to suit any specific use-case.

Since the MGP output is directly multiplied by weights $c_i$, the classification model can be interpreted as a scoring mechanism where each past point $y_{\mathrm{MC},ij}$ contributes $\alpha_{i,0}\beta_{ij,0}$ to the time series being classified as positive, and $\alpha_{i,1}\beta_{ij,1}$ to the time series being classified as negative. The positive and negative scores are then normalised to represent probabilities of the positive or negative labelling. As we design both $\boldsymbol{\alpha}$ and $\boldsymbol{\beta}$ to be non-negative, we can hence directly look at the average $\boldsymbol{\alpha}$ and $\boldsymbol{\beta}$ over Monte Carlo samples to see which time points and features contribute most strongly to the network's positive or negative decision.

## 4 DATA

Sepsis is defined as a life-threatening organ dysfunction caused by a dysregulated host response to infection (Singer et al., 2016). The latter is usually interpreted as the administration of antibiotics coupled with the culture of blood samples, generating a suspicion of infection window, whereas the former is interpreted as a two point increase in Sequential Organ Failure Assessment (SOFA) within such a suspected infection window. We make use of the MIMIC-III data set (Johnson et al., 2016) and encode the Sepsis-3 criteria following Johnson & Pollard (2018)'s code available on GitHub, with the help of Moor et al. (2019)'s code that the authors have generously provided.

One key difference between our assumptions and Moor et al. (2019)'s is the handling of missing SOFA contributor values: if one or more SOFA contributors are missing, Moor et al. do not calculate the total score. On the other hand, we assume such a contributor to be within a healthy norm, implying a zero contribution. The latter heuristics is in line with the official Sepsis 3 definition in Singer et al. (2016). In order to validate our results, we carry out all experiments using both labelling techniques.

We proceed to extract times series of case and control patients for a set of commonly recorded vitals, labs and static variables and normalise their values. Following Moor et al. (2019), in order to keep the data set length balanced, we match the time series lengths of control patients to those of case patients using the class balance ratio. In addition, we create up to seven copies of each time series and truncate the last zero to six hours of data, effectively creating early prediction patients and augmenting our data set. We remove excessively noisy or computationally intensive data and train the model over

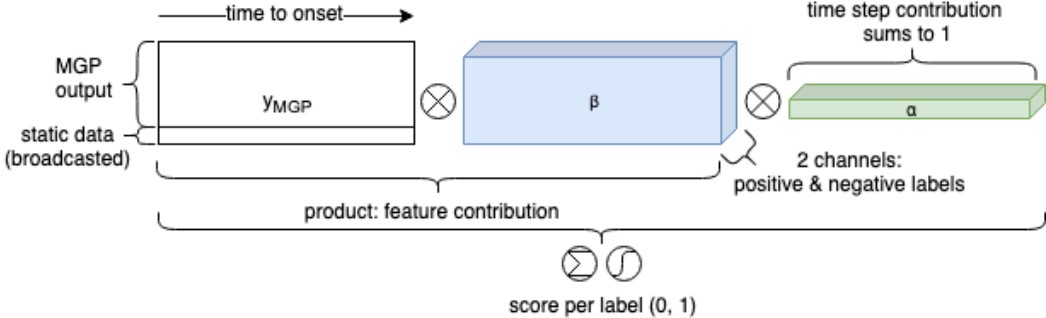

Figure 2: Interpretation of the different attention weights in our model.

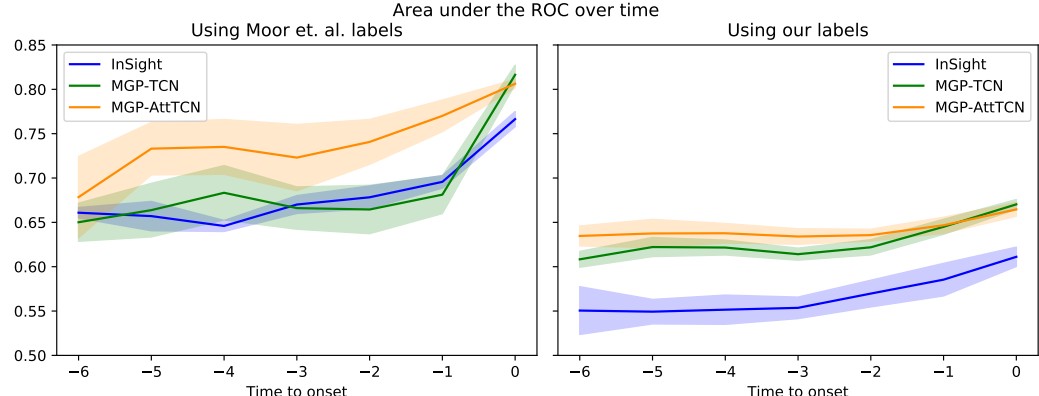

Figure 3: Performance of different models. It can be seen that our proposed labels are harder to fit than the ones by Moor et al. (2019). Moreover, our proposed model outperforms the baselines on both label sets, especially for earlier prediction horizons.

different hyperparameters, randomly resampling an equal number of case and control patients to counteract the data set imbalance.

## 5 EXPERIMENTAL RESULTS

We compare our model's performance to the performance of the InSight algorithm (Calvert et al., 2016) and to the state-of-the-art MGP-TCN algorithm (Moor et al., 2019). Figure 3 shows the predictive performance of the models for different time horizons.

### 5.1 COMPARISON BETWEEN DIFFERENT DATA LABELS

The first result is the difference in performance of models applied to the different labelling methods. The SOFA contributor assumption Moor et al. make has two main implications. Firstly, it considerably restricts the number of patients. Assuming that sicker patients receive more medical attention, the patients included are likely to be in worse conditions than the septic patients excluded and hence easier to classify. Secondly, it delays sepsis onset. For example, a patient having a severe liver failure with few other recorded vitals, followed by an overall collapse further in time will have septic onset at the time of its liver failure in our records, whereas it will only be considered septic at the time of the overall collapse in Moor et al.'s labels. On the other hand, the labels we produce reflect the incomplete nature of medical data: even if only a part of all the potentially relevant tests are carried out at any given time, a doctor must be able to assess a patient's well-being and foresee potential complications. The difference in labels implies a discrepancy in task difficulty: Moor et al.'s labels present an easier learning problem, however they define a more narrow use-case in real-world scenarios.

Indeed, when assessing the performance of the different models on the two different data labellings, it becomes evident that our proposed labels are harder to fit. This means that predicting sepsis in a realistic setting on the intensive care unit is probably much harder than previous work would suggest.

### 5.2 MODEL PERFORMANCE

We find that our MGP-AttTCN model has a better performance when presented with patients further in time from sepsis onset. In the case of Moor et al.'s labels the difference is clearly noticeable, whereas with our labels it is of lower statistical significance. With our labels, both MGP-TCN and MGP-AttTCN have a stronger performance than InSight. The intuition behind this result is the robustness of the models to missing data: both MGP-TCN and MGP-AttTCN account for the data uncertainty and hence have a better performance on lower resolution and more irregular data.

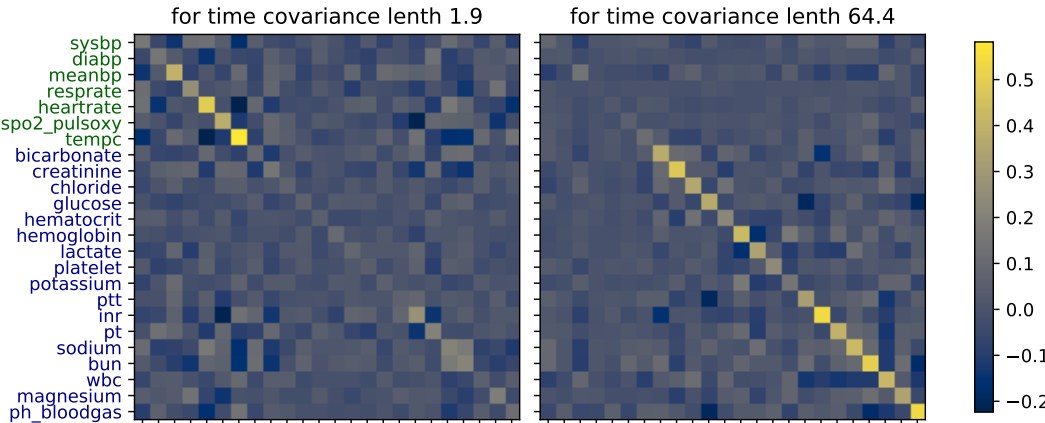

Figure 4: Heatmaps of the learned MGP covariance matrices between the data features for the two different smoothness clusters.

On Moor et al.'s labels, the MGP-TCN model does not seem to significantly outperform the InSight model, suggesting that those labels might be easy enough to not require a particularly pronounced robustness to missing data. However, the additional attention of the proposed MGP-AttTCN model does seem to gain a clearer advantage here than on our labels, presumably due to a more complete set of features that can be attended to.

### 5.3   MGP INTERPRETABILITY

Inspecting the learned covariances (Figure 4), we notice that the two OU lengthscales converged to represent two clusters within the selected variables: a shorter lengthscale (around two hours) represents noisy data, whereas a larger lengthscale (around 64 hours) represents smoother observations. In addition, the feature covariance matrix for the short lengthscale puts more emphasis on vitals, while the one for the long lengthscale puts more emphasis on labs, fitting our initial intuition that vitals vary more rapidly. Graphically, one can observe this by inspecting the diagonals on the covariance heatmaps.

On a more granular level, the two covariance matrices also provide insights about the underlying variables. One can for instance observe that the body temperature (*tempc*) has a larger variance than the systolic and diastolic blood pressure (*sysbp*, *diabp*), following the general clinical intuition. Moreover, we can observe correlations between different features, such as a negative correlation between temperature and heart rate, which also seems to coincide with the general medical expectation. These covariances can then for instance be used by the model to extrapolate a full function from a single INR observation with an inverse correlation to the pulse oximetry observations (Fig. 5).

### 5.4   ATTENTION WEIGHTS

One important benefit of our model compared to current approaches is its interpretability due to the attention mechanism. Once the samples have been drawn, the weights $\alpha$ and $\beta$ provide us with more information about the importance of different time points and features for the model's behaviour. The attention weights for an exemplary patient trajectory are depicted in Figure 5.

Overall, the absolute values of $\alpha$ are small for points further from the prediction time and increasingly larger closer to it. A good example of this behaviour is the fourth row in Figure 5, where feature importance increases in time. We can also see there, that different features can have opposing effects on the prediction. While the elevated heart rate close to the prediction time increases the likelihood of a sepsis prediction (first column, yellow weights), the lowered prothrombin values reduce this likelihood (third column, blue weights). Interestingly, the low prothrombin values are not actually measured in this example, but predicted by the MGP purely based on the other measured features and the learned covariances.

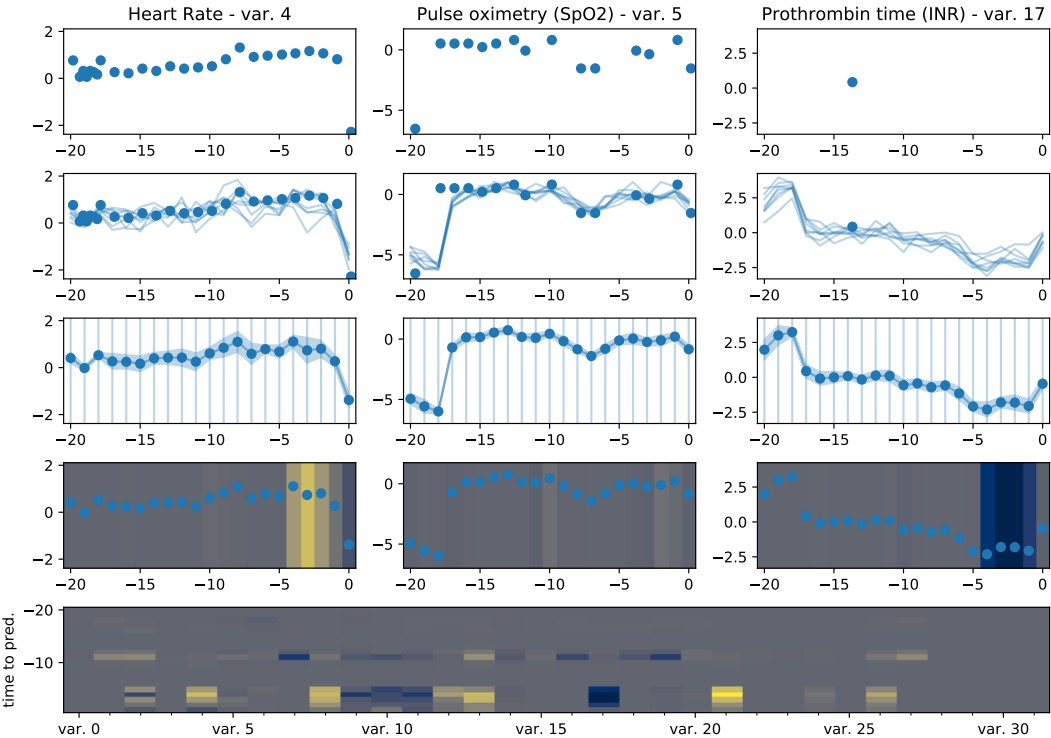

Figure 5: Visualization of the journey of an exemplary patient trajectory through our proposed model architecture. The raw features (row 1), measured at irregular time points, are interpolated by an MGP (row 2). Samples from the MGP posterior can then be aggregated into means and variances for each feature on a fixed, regularly-spaced time grid (row 3). These values are then attended to by the TCN (row 4), where positive attention weights are yellow and negative ones blue. Row 5 shows the attention weights separated by features (x-axis) and time point (y-axis).

Finally, $\alpha \times \beta \times y_{\text{MC}}$ gives the individual score contribution of each feature at each time point. These weights are shown in the last row of the figure. It can again be seen that the attention weights are generally larger in magnitude closer to the prediction time. Moreover, about half of the features have significant non-zero attention weights, while the others seem to not be important for the prediction in this example.

These visualizations could be used by doctors to make an informed decision about whether or not to trust the prediction of the model for each given patient, thus facilitating the interpretability and accountability that is crucial in medical applications.

## 6 CONCLUSION

We have shown that current data sets for the early prediction of sepsis underestimate the true difficulty of the problem and proposed a new labelling for the MIMIC-III data set that corresponds more closely to a realistic intensive care setting. Moreover, we have proposed a new machine learning model, the MGP-AttTCN, which outperforms the state-of-the-art approaches on the easier labels from the literature as well as on our proposed harder labels. Additionally, our model provides an interpretable attention mechanism that would allow clinicians to make more informed decisions about trusting its predictions on a case-by-case basis.

Potential avenues for future work include a more thorough discussion with clinicians to potentially make our proposed labels even more representative of the real-world task, and architectural improvements, for instance by meta-learning the MGP prior (Fortuin & Rätsch, 2019), amortizing the latent MGP inference for performance gains (Fortuin et al., 2019), or discretizing the latent space for improved interpretability (Fortuin et al., 2018).

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

## APPENDIX A  DATA PROCESSING

### A.1  DATA LABELLING

Please refer to *link omitted for anonymity* for more details how we derived the MIMIC-III sepsis labels.

### A.2  DATA EXTRACTION

**Patient Inclusion**  We filter for patients admitted to Intensive Care Units (ICU) who are more than 14 years old and with valid records. Case patients are patients having sepsis onset within their ICU stay, whereas control patients have not developed sepsis nor have an ICD discharge code referring to sepsis. Starting with 58'976 patients, we find 14'071 control patients and 7'936 case patients using our labels, versus 1'797 cases using Moor et al. labels.

**Feature extraction**  Reviewing sepsis related literature and commonly extracted laboratory and vital recordings, we extracted all features which were reported at least once for more than 75% of the included population. The final 24 dynamic features are reported in Table 3. We also extracted static features - age, gender and first ICU admission department.

**Case-control matching**  As the goal is to predict sepsis prior to onset, the cases data was extracted between ICU admission and sepsis onset. Note that sepsis onset happens early within ICU admission, with the median patient getting sick at 3.4 hours of admission. On the other hand, patients not developing sepsis are more likely to recover completely, and do so in a lengthier time frame. In addition, once they are close to discharge, their vitals and labs are within the norms. Hence, both the length and the values of the time series are strong discriminatory factors which ease the classification. We hence carry out a matching strategy similar to Moor et al. (2019): following the class imbalance ratio, we associate each control time series to a case time series and truncate the control to have the same length as the case from ICU admission. We then discard patients with less than 40 data points within the selected window, and - for computational tractability - truncate the first $N_p - 250$ initial values of patients' time series in order to keep a maximum of 250 data points per patient.

**Horizon augmentation**  As our goal is to predict sepsis early, we augment the data by creating new shorter time series. For each time series, we create six copies, where each copy represents a different horizon to onset. We then proceed to truncate the last one to six hours prior to onset from the time series copies. In order to keep data consistency, we once again discard time series with less than 40 observations. In Tables 1 and 2 we illustrate the data distribution per horizon.

Table 1: Augmented Dataset Description with Moor et al. labels

| Horizon to onset | N. of patients | N. of obs. per patient |
|:---:|:---:|:---:|
| 0 h | 15'123 | $69.9 \pm 59.6$ |
| 1 h | 11'258 | $56.6 \pm 59.1$ |
| 2 h | 8'478 | $61.4 \pm 62.8$ |
| 3 h | 6'554 | $66.5 \pm 65.9$ |
| 4 h | 5'233 | $70.6 \pm 69.0$ |
| 5 h | 4'162 | $76.3 \pm 71.9$ |
| 6 h | 3'390 | $81.9 \pm 74.3$ |

**Data split**  Finally we split the data into training, validation and testing sets, respectively capturing 80%, 10% and 10% of the data. We then normalise the data by subtracting the training set mean and dividing by the training set standard deviation of each feature.

Table 2: Augmented Dataset Description with authors labels

| Horizon to onset | N. of patients | N. of obs. per patient |
|:---:|:---:|:---:|
| 0 h | 20'075 | $64.0 \pm 65.5$ |
| 1 h | 15'832 | $62.5 \pm 65.6$ |
| 2 h | 12'080 | $66.1 \pm 67.2$ |
| 3 h | 9'441 | $69.7 \pm 68.1$ |
| 4 h | 7'484 | $73.4 \pm 68.5$ |
| 5 h | 6'007 | $77.1 \pm 68.2$ |
| 6 h | 4'876 | $81.2 \pm 67.4$ |

Table 3: List of dynamic features

| Vitals | Labs | |
|:---:|:---:|:---:|
| Sys. blood pressure | Bicarbonate | PTT |
| Dia. blood pressure | Creatinine | INR |
| Mean blood pressure | Chloride | PT |
| Resp. rate | Glucose | Sodium |
| Heart rate | Hematocrit | BUN |
| SpO2 pulse ox. | Hemoglobin | WBC |
| Temperature (C) | Lactate | Magnesium |
| | Platelet | pH blood gas |
| | Potassium | |

## APPENDIX B  BASELINES

### B.1  DATA PREPARATION

In order to benchmark our MGP model, we build some baselines homogenising the data sampling. For each hour and variable, we take the average of the available observations. If a given hour has no observations, we carry forward the average of the previous hour. In this manner, we generate an hourly sampled time series for each patient. We then proceed to normalise the size of each patient matrix by setting a time window of observation $N$. For patients having more than $N$ observations $N_p$, we discard the first $N - N_p$ observation; whereas for patients having less than $N$ observations we pad the beginning of the matrix with zeros.

$$
\begin{pmatrix} y_{p,1,t_1} & \cdots & y_{p,1,t_{N_p}} \\ \vdots & \ddots & \\ y_{p,M,t_1} & \cdots & y_{p,M,t_{N_p}} \end{pmatrix} \xrightarrow{\text{carry forward}} \begin{pmatrix} y_{p,1,1} & \cdots & y_{p,1,N_p} \\ \vdots & \ddots & \\ y_{p,M,1} & \cdots & y_{p,M,N_p} \end{pmatrix} \tag{14}
$$

$$
\xrightarrow{\text{normalise}} \begin{cases} \begin{pmatrix} y_{p,1,N-N_p} & \cdots & y_{p,1,N} \\ \vdots & \ddots & \\ y_{p,M,N-N_p} & \cdots & y_{p,M,N} \end{pmatrix} & \text{if } N_p \geq N \\ \begin{pmatrix} 0 & \cdots & 0 & y_{p,1,N-N_p} & \cdots & y_{p,1,N} \\ \vdots & & \vdots & \vdots & \ddots & \\ 0 & \cdots & 0 & y_{p,M,N-N_p} & \cdots & y_{p,M,N} \end{pmatrix} & \text{oth.} \end{cases}
$$
$$\tag{15}$$

We choose to align the end of the time series as opposed to the beginning as the relative importance of time points is to when a patient becomes sick rather to when he is admitted to the ICU.

As a next step, we augment the data to focus on different time series in a similar manner than for irregularly sampled data. We create seven copies of each time series, for each copy we discard the last zero to six hours, then normalise the matrix as above. We hence generate a dataset $\mathbf{Y}_{\text{BL}} = \{Y\}_q = \{\{y_{q,ij}\}_{i,j=1}^{N,M}\}_q$ where $q$ represent all augmented the time series.

## B.2 InSight

The InSight scoring model is one of the few machine learning algorithms to surpass the proof-of-concept stage with multiple research, economic and clinical trials (Calvert et al., 2016; Desautels et al., 2016; Calvert et al., 2017; Mao et al., 2018). We therefore include it as a baseline to our model.

The key concept of the model is to use few largely available vitals, build some handcrafted features and train a simple classification model.

Here is an account of our interpretation of the author's method. The features extracted are based on a six consecutive hour window. For each six hour window, we extract each variable's mean $M_i$ and difference $D_i$ (last observation minus first observation) over the window. We also extract variables pairs correlation $D_{ij}$ and *triplet correlation* $D_{ijk}$; where $i, j, k$ are observed variables. We interpret the latter as a relaxation of the Pearson correlation: if the correlation between two variables is

$$\rho_{XY} = \frac{\mathbb{E}[(X - \mu_X)(Y - \mu_Y)]}{\sigma_X \sigma_Y} \tag{16}$$

then we define the *triplet correlation* as

$$\rho_{XYZ} = \frac{\mathbb{E}[(X - \mu_X)(Y - \mu_Y)(Z - \mu_Z)]}{\sigma_X \sigma_Y \sigma_Z} \tag{17}$$

We then classify the difference and correlations as either positive, negligible or negative using their distribution quantiles over every patient and six hour window observed. Note that given the high level of data missingness, many variables are calculated by forward imputation and hence have no variance over six hours. To adjust for the high number of zero correlations, we calcualte the quantiles of non-zero correlations and define:

$$\hat{D}_i = \begin{cases} 1 & \text{if } D_i > q^*(2/3) \\ -1 & \text{if } D_i < q^*(1/3) \\ 0 & \text{otherwise} \end{cases} \tag{18}$$

where $q^*$ is the adjusted quantile function. We proceed in a similar manner for the correlations and triplet correlations.

In order to keep the results comparable to the AttTCN fixed window $N$, we extract $N - (6 - 1)$ six consecutive hour window and vectorise the resulting features, generating in total

$$n_{\text{features}} = \left(N - 5\right) \times \left(2 \times M + \binom{M}{2} + \binom{M}{3}\right) \tag{19}$$

features per patient.

Although the original paper does not specify which classification method the authors employ, we derive by their description of a *dimensionless score* that the method is a logistic regression.

## APPENDIX C  MODEL DETAILS

### C.1 TCN PROPERTIES

TCNs are a class of neural networks composed of *causal* convolutions stacked into Residual Blocks. A causal convolution is a 1D convolutional layer which only takes inputs from the past to generate its output, avoiding any data leakage. Residual Blocks are made of two causal convolutional layers together with ReLU activation functions, dropout and L2 regularisations.

### C.2 STATIC VARIABLES

In our model implementation, we decide to integrate the static variables to the MGP output. Another choice we considered is to integrate the data to the output of the attention model. Once the weights $\boldsymbol{\alpha}$ and $\boldsymbol{\beta}$ have been created, we can introduce a *bias* term to

$$\hat{\mathbf{l}}_i = \text{Softmax}\left(\sum_m^N \sum_m^{M+Q} \mathbf{c}_{i,nm} + \mathbf{b}_{\text{static}}\right) \in [0, 1]^2 \tag{20}$$

Table 4: Hyperparameter search

| Hyperparameter | Random Search | |
|---|---|---|
| | min | max |
| MGP Monte Carlo samples | 4 | 20 |
| TCN kernel size | 2 | 6 |
| TCN number of Residual Blocks | 2 | 12 |
| TCN number of hidden layers | 10 | 55 |
| TCN dropout rate | 0 | 0.99 |
| TCN L2 regularisation | 0 | 250 |

Table 5: Area under the ROC curve for Moor et. al. labels

| Time to onset | InSight | MGP-TCN | MGP-AttTCN |
|---|---|---|---|
| 6h | $66.4 \pm 0.7$ | $65.0 \pm 2.2$ | $\mathbf{67.0 \pm 4.3}$ |
| 5h | $65.9 \pm 1.5$ | $66.4 \pm 2.8$ | $\mathbf{73.2 \pm 2.7}$ |
| 4h | $64.7 \pm 0.6$ | $68.3 \pm 2.9$ | $\mathbf{73.4 \pm 2.9}$ |
| 3h | $67.0 \pm 1.0$ | $66.6 \pm 2.4$ | $\mathbf{72.2 \pm 3.6}$ |
| 2h | $67.7 \pm 1.2$ | $66.4 \pm 2.7$ | $\mathbf{73.9 \pm 2.5}$ |
| 1h | $69.5 \pm 0.8$ | $68.2 \pm 2.0$ | $\mathbf{76.8 \pm 1.7}$ |
| 0h | $76.4 \pm 0.9$ | $\mathbf{81.4 \pm 0.9}$ | $80.5 \pm 0.5$ |

where $\mathbf{b}_{\text{static}}$ is generated by the output of a two layers multilayer perceptron applied to the static data. Although this solution is computationally lighter and provides a more elegant interpretation, it does not allow the attention mechanism to utilise the information about the patient's static state when making a decision about its vitals and labs values. For the scope of this paper, we hence decided for earlier static data integration.

## C.3 PARAMETER HYPERSEARCH AND TRAINING

As the datasets are highly unbalanced, we carry out a case set oversampling: we randomly resample the case set to have the same size as the control set. In addition, at each iteration we sample equally the same number of cases and controls, then feed a shuffled version into the model. In this manner the model will see an equal number of controls and cases and will not become biased towards zero labels. This procedure does not happen for either of the validations and test sets, as the results would not compare to real life settings.

For both our core model MGP-AttTCN and all baselines, in order to select the best possible hyperparameters, we performed a hyperparameter random search, as described in Table 4.

## C.4 NUMERICAL RESULTS

The results are shown in Tables 5 and 6.

Table 6: Area under the ROC curve for our labels

| Time to onset | InSight | MGP-TCN | MGP-AttTCN |
|---|---|---|---|
| 6h | $55.5 \pm 2.7$ | $60.9 \pm 0.9$ | $\mathbf{63.6 \pm 0.9}$ |
| 5h | $55.5 \pm 1.4$ | $62.4 \pm 1.0$ | $\mathbf{63.9 \pm 1.6}$ |
| 4h | $55.5 \pm 1.6$ | $62.4 \pm 0.9$ | $\mathbf{64.1 \pm 1.2}$ |
| 3h | $55.9 \pm 1.3$ | $61.6 \pm 0.7$ | $\mathbf{63.6 \pm 0.9}$ |
| 2h | $57.3 \pm 1.5$ | $62.4 \pm 0.9$ | $\mathbf{63.6 \pm 0.6}$ |
| 1h | $58.8 \pm 1.8$ | $64.5 \pm 0.9$ | $\mathbf{64.7 \pm 0.9}$ |
| 0h | $61.4 \pm 1.1$ | $\mathbf{67.0 \pm 0.6}$ | $66.5 \pm 0.9$ |

