# OpenReview forum: "MGP-AttTCN: An Interpretable Machine Learning Model for the Prediction of Sepsis"
_ICLR.cc/2020/Conference — Reject_

### Official Review · AnonReviewer1 · 2019-10-19
**Official Blind Review #1**

**Rating:** 3

**Review:**

I've read the rebuttal and I'd like to keep my score as is. My main concern is the questionable role of attention in making the model more interpretable (which is the main contribution of the paper).

###########################

The paper proposes a new model for automated sepsis detection using multitask GP and attention-based GP. The sepsis detection problem is of paramount importance in the clinical domain and the authors rightly emphasized that point. Also the paper tries to combine interpretability with prediction accuracy by using attention mechanism.

The paper is generally well written and well motivated; however, in terms of technical novelty and empirical evidence the paper can be further improved.

The MGP-AttnTCN model is mostly a minor modification of the model proposed by Moor et al. 2019 and has the additional attention element to be more interpretable. Unfortunately, it’s not easy for an ICLR reader without any medical background to evaluate the validity of the interpretability results provided in the paper. Furthermore, the recent works in NLP have agued against the value of attention for interpretability (see for instance “Attention is not Explanation” by Jain & Wallace 2019). That said, I believe the paper is probably a better fit for a machine learning in healthcare venue (such as MLHC).

In terms of empirical evidence of the prediction accuracy the paper only compares with Moor et al 2019 (which does not show a significant improvement in the realistic setting) and a much older InSight paper (2016). This would have been typically enough for a paper with major technical novelty; however, for this paper, I believe adding more recent baselines and discussing the advantages of the method over these baselines would be necessary.

Minor:
Caption in Figure 1 can be more informative and useful for the reader if you add more details on different parts of the model.
“Graphically, once can observe” should be “one can observe” .

**Experience Assessment:**

I have read many papers in this area.

**Review Assessment: Checking Correctness Of Derivations And Theory:**

I assessed the sensibility of the derivations and theory.

**Review Assessment: Checking Correctness Of Experiments:**

I assessed the sensibility of the experiments.

**Review Assessment: Thoroughness In Paper Reading:**

I read the paper at least twice and used my best judgement in assessing the paper.

---

> ### Author Response · Authors · 2019-11-13
> **Thank you for your review**
>
> Thank you for taking the time to read through our paper and share your thoughts. We are going to respond to your comments in the following.
>
> Interpretability:
> An approach into the validation of interpretability of the results is given in Figure 5: the time-points closer to the sepsis onset are deemed more important by the model, which is intuitively where you would see a patient’s health worsening.
> As per the relevance of the attention mechanism, Wiegreffe and Pinter argued against the point made by Jain and Wallace (“Attention is not not Explanation”). We believe a fully fleshed proof of the explainability of attention is beyond the scope of this paper.
>
> Baselines:
> Regarding comparison to baselines, to the best of our knowledge, the work of Moor et al. is the state of the art. Moreover, we decided to also compare our work to Insight, as this interpretable model is the most advanced one validated by the clinical community (as it is now undergoing clinical trials). Our improvement over Moor et al. is not only in the actual performance (at least on their labels) but mostly in the interpretability of the model.
> Minor:
> We updated Figure 1 to make the caption more useful.
>
> If you have any further suggestions on how to improve the paper, please let us know.

---

### Official Review · AnonReviewer3 · 2019-10-26
**Official Blind Review #3**

**Rating:** 1

**Review:**

The authors present a Sepsis-3 compliant labeling of the MIMIC-III dataset and a sepsis-prediction model largely based on MGP-TCN that uses attention mechanisms to enable explainability.

It is not entirely clear what the authors mean by MC samples from Y_MGP, are these simply samples from the posterior in (6)?

If z_j and z'_j are M-dimensional, how does one apply (8) and (9) for W_{\alpha,0}, W_{\alpha,1} being (M+Q)-dimensional or W_{\beta,0}, W_{\beta,1} being matrices?

The labelling of the data, largely following Johnson & Pollard (2018) and Moor et al (2019), is only different to Moor et al (2019) in the assumption that in the SOFA calculation missing values have zero contribution. Unless the authors provide evidence that this is reasonable, it is not necessarily clear whether labels resulting from the proposed scheme will be biased and affected by differences in clinical practice at different sites or data collection practices. That being said, it is not clear whether the proposed labeling is a contribution from the work.

The fact that the proposed labels are harder to fit does not imply that the proposed labels are better or more reasonable. This provided that is difficult to know (without ground truth) whether the difficulty originates from a broader use case (not as easy as Moor et al (2019)) or labels being noisy, imperfect proxies for sepsis diagnosis. I understand the author's motivation for doing it, however, their approach is not sufficiently justified. I also agree that predicting sepsis in a realistic setting is harder than suggested in prior work, however, the proposed labeling does not necessarily yields evidence of that being the case.

The interpretation of the covariance matrices of the MGP is interesting, though not surprising considering that covariates in green are measured regularly while blue covariates are ordered sparingly.

Figure 5 is interesting, though raises questions about of interpretability of the model. How should unobserved covariates be interpreted (INR in Figure 5)?

In summary, the contributions of the present work are not sufficiently justified (labeling), the novelty of the proposed model is minor, relative to MGP-TCN, and the added value of the attention mechanism as a means to interpret predictions in terms of the journey of a patient is not clear.

Minor:
- Figure 1 needs a better caption. Being in page 2 makes it very difficult to understand.
- TCN is used before being defined.
- In (1) it should be t_{p,i,k} not t_{p,k,i}

**Experience Assessment:**

I have published one or two papers in this area.

**Review Assessment: Checking Correctness Of Derivations And Theory:**

I carefully checked the derivations and theory.

**Review Assessment: Checking Correctness Of Experiments:**

I carefully checked the experiments.

**Review Assessment: Thoroughness In Paper Reading:**

I read the paper thoroughly.

---

> ### Author Response · Authors · 2019-11-13
> **Thank you for your review**
>
> Thank you for taking the time to read through our paper and share your thoughts. We are going to respond to your comments in the following.
>
> New sepsis labels:
> Regarding the different labelling of the data, we will make the code available once the paper will be reviewed - as it is now on a (non anonymous) github repository. As such, the labelling is part of the contributions of the paper.
> The decision to have zero contributions reflects that we assume a patient is healthy unless proven otherwise. This is in line with the official Sepsis-3 guidelines, which claim that “the baseline SOFA score can be assumed to be zero in patients not known to have preexisting organ dysfunction.” (https://jamanetwork.com/journals/jama/fullarticle/2492881, Box 3). Our assumption is that the authors of the Sepsis-3 guidelines are aware of possible clinical practices and have taken them and other clinical biases into consideration when fleshing out their recommendations.
> Moreover, as opposed to Moor et al who ignore cases that do not have complete data, our labels will also contain patients with fewer records and hence ‘noisier’ time series, but also a broader and more realistic use-case. On the other hand, only looking at well documented patients is not in line with the aim of the research stream: if a patient is already well attended, then doctors are already well aware of their health conditions and a diagnostic support tool would only bring marginal benefit.
>
> Interpolation in Figure 5:
> Your point on Figure 5 can be explained by the Multitask nature of the Gaussian Process: even if there is no input for that specific covariate, the model is able to infer its value from the other values it is able to record.
>
> MGP samples:
> Regarding the MC samples y_{MGP}, as written in the original paper, they are taken from the posterior over t’ defined by \mu and \Sigma in equation (6) in order to approximate its distribution. Could you be more specific about which part is unclear?
>
> Dimensionality of the latents:
> z_j and z_j’ should be N x (M+Q), we amended the paper. Thank you for spotting that.
>
> If you have any further suggestions on how to improve the paper, please let us know.

---

### Official Review · AnonReviewer2 · 2019-10-30
**Official Blind Review #2**

**Rating:** 8

**Review:**

The authors consider a combination of an Gaussian model and neural network learned together to be able to find specific Gaussian features that would predict sepsis in a more intuitive, interpretable way for a medical experts. Also, they have provided a new labelling for the MIMIC-III dataset, which is of great value.

Usually constraining the feature space reduces the accuracy, as one tends to miss important features. However, here, one is using a Gaussian model to generate a feature space that is easier to train with a neural network (by filling the sparse data by Gaussian process interpolation) but also augment the dataset.

The author report that his improves prediction. Unfortunately, I did not find tables of solutions where one could the actual impact. The authors should include a numerical table of their result comparison results.  Now there is only a narrative in section 5.3 and an image showing that at different covariance times, different feature groups starts to interpret the results. Obviously, a question arises if the model would perform even better with a combination of covariance times, or is there some covariance time range that is missing that would improve the result even more.

The Gaussian model creates smooth interpolation of data spaces and also forces the training to look at corresponding smoothened features - that are very good for human eyes. However, there are situations (like the detecting heart beat from a video from a head moving with a recoil from the blood rushing to the brain) in where the signal is too weak for human to see, but is definitely there for a computer.  I would state that this as interpretable,  as an explicitly visible signal would be. Even  shorter time constant signal might be valuable as well, but it would not be visible here...  It seems that in sepsis, it was a good idea, as it improves the result compared to the situation of not using the Gaussian model.

One has to be careful to state that this would address the interpretability of the results. Gaussian process by itself is not giving understanding nor interpretability as it is too general. But it can make the provided solution "teachable" to a human expert by showing what visible features one can track.

Compare this to a situation where one is using physical model to regularize detection. It has the analogous two model structure like the one in the manuscript. In https://xbpeng.github.io/projects/SFV/index.html the authors of the paper report that that one can achieve a better pose estimation by constraining the pose to only those that are achievable by a physical model based policy trained by reinforcement learning. This one is able to "interpret" the pose.

As a summary, the authors have done solid and valuable work in improving the accuracy detection. They should have a more formal way to present the results and baseline comparisions as tables. On the explainability and interpretability, there remains a lot of interpretations and one has lots of explaining to do, even after this manuscript.






**Experience Assessment:**

I have read many papers in this area.

**Review Assessment: Checking Correctness Of Derivations And Theory:**

I assessed the sensibility of the derivations and theory.

**Review Assessment: Checking Correctness Of Experiments:**

I assessed the sensibility of the experiments.

**Review Assessment: Thoroughness In Paper Reading:**

I read the paper thoroughly.

---

> ### Author Response · Authors · 2019-11-13
> **Thank you for your review**
>
> Thank you for taking the time to read through our paper and share your thoughts. We are going to respond to your comments in the following.
>
> Numerical performance results:
> We added the table with the numerical results in Appendix C.4.
>
> Choice of covariance times:
> In response to the time covariances, we initially started with one covariance for all variables, which performed worse than the presented model. We also clustered the different features solely based on data sampling frequency, and found that two clusters were optimal: the clusters have low wss and are intuitive to the medical staff. It is especially given the latter point that we decided for two clusters instead of treating the number of covariances as a hyperparameter.
>
> Interpolation:
> As per the interpolation of the signal, we would like to point out that even the most frequently sampled data is sampled every 15 minutes. As such, micro-movements in data would already be missing. However, in order to limit the amount of smoothing, we decided to use a kernel that has no moments greater than two. As such, these kernels are able to capture ‘jumpier’ behaviours.
>
> If you have any further suggestions on how to improve the paper, please let us know.

---

### Decision · Program_Chairs · 2019-12-19

**Decision:**

Reject

**Comment:**

The problem of introducing interpretability into sepsis prediction frameworks is one that I find a very important contribution, and I personally like the ideas presented in this paper. However, there are two reviewers, who have experience at the boundary of ML and HC, who are flagging this paper as currently not focusing on the technical novelty, and explaining the HC application enough to be appreciated by the ICLR audience. As such my recommendation is to edit the exposition so that it more appropriate for a general ML audience, or to submit it to an ML for HC meeting. Great work, and I hope it finds the right audience/focus soon.